# A Design for the High Yield Photoneutron Source Target Station

**DOI:** 10.3390/ma15217674

**Published:** 2022-11-01

**Authors:** Yuxuan Lai, Yigang Yang

**Affiliations:** 1Department of Engineering Physics, Tsinghua University, Beijing 100084, China; 2Key Laboratory of Particle & Radiation Imaging, Tsinghua University, Ministry of Education, Beijing 100084, China

**Keywords:** photoneutron source, high neutron yield, low energy accelerator, target station, hadron-based method

## Abstract

Low energy accelerator driven neutron sources are promising candidates to obtain a neutron yield as high as 10^14^ n/s, which is required for a variety of applications, such as boron neutron capture therapy, neutron imaging, and neutron scattering. The methods to generate neutrons can be divided into two categories: hadron-based and photon-based methods. In order to better understand which kind of source would be the better choice for delivering a brilliant neutron beam robustly, in this paper, the underlying principles of neutron production, as well as the simulation results of neutron yield, target heat dissipation, thermal stress, and reaction byproducts concentration of these two types of neutron sources, will be elaborated on. A preliminary photoneutron target station design based on a 50 MeV/50 kW electron linear accelerator, including the optimized neutron yield, thermal hydraulic analysis, and shielding calculation, is presented as well to demonstrate the method to deliver brilliant thermal neutron beam of 1.03 × 10^10^ cm^−2^ s^−1^ sr^−1^.

## 1. Introduction

Since the discovery of the neutron by James Chadwick in 1932 [1], neutron related technologies have come to play irreplaceable roles in many different fields, such as neutron scattering in the areas of physical, chemical, and material science [2], neutron imaging the industrial applications [3], and boron neutron capture therapy (BNCT) in the medical applications [4]. Due to the low probabilities accompanying the neutrons’ moderation and detection process, however, the practical applications of neutron inherently demand that neutron sources should deliver intense neutron beams to meet the requirements of counting statistics. For example, a BNCT system requires a higher than 10^14^ n/s neutron yield of the neutron source to generate epithermal neutrons with flux higher than 10^9^ cm^−2^ s^−1^ for an effective treatment [5]. A thermal neutron imaging system, with a typical aspect ratio of 100:1, demands a neutron yield of 10^14^ n/s, when the efficiency of moderating fast neutrons to thermal neutrons is 10% and the required thermal neutron flux at the position of the detector is about 10^6^ n/cm^2^/s, in order to complete a neutron imaging within an acceptable temporal duration (for example, several minutes). Therefore, a brilliant neutron source would be anticipated to provide neutrons with a yield higher than 10^14^ n/s before it could be deemed a practical source that can be deployed for a variety of applications.

Although both the reactor neutron sources and the spallation neutron sources, which are the most brilliant sources, can produce neutrons with the yield of higher than 10^16^ n/s, their practical applications are unfortunately hindered due to their large footprint, high constructing and operating cost, and the scarcity of the beam time, so we must have our eyes on the other neutron sources. The isotopic neutron sources are ruled out for their low neutron yields, and the neutron generators are also not practically considered due to their quite short life-span that is typically smaller than several thousand hours, as well as the low neutron yield, although the neutron yield of 10^12^ n/s has been reported by colleagues [6]. Consequently, neutron sources driven by the low energy accelerators, where the kinetic energy of accelerated charged particles is usually not high enough to induce spallation reactions, perhaps would be the only candidate to facilitate the applications of neutron technologies in the areas of industrial inspection and medical treatment.

Neutrons are stable in the potential well formed by other nucleons of the target nuclei, with a typical well depth of 7~8 MeV. In order to liberate neutrons, enough energy should be fed into the target nuclei. Depending on the force related to this energy, the methods to produce neutrons can be categorized into two methods: hadron-based and photon-based methods. In the hadron-based method, proton, deuteron, or even heavier nuclei would be accelerated to MeV or even higher energy to penetrate the coulomb barrier formed by the target and the incident particle, in order to form a compound state with the excited energy, which is equal to the summation of the relative kinetic energy of the incident nucleus and the binding energy between the two nuclei. In the photon-based method, electrons are usually used to produce energetic bremsstrahlung photons, whose energy is absorbed by the target nuclei via the giant dipole resonance (GDR) or quasi deuteron (QD) mechanisms to subsequently emit neutrons [7]. Both the nuclear force and the electromagnetic force are involved in the hadron-based method and only electromagnetic force is involved in the photon-based method. The different interaction behaviors of hadrons and photons (as well as their precursors, electrons) in the matter lead to varied performances in the neutron production, heat dissipation, hydrogen/helium embrittlement, and other issues that may affect the evaluation of the feasibility of the two kinds of sources to be deployed in the practical applications.

In this study, the underlying principles of the neutron production, as well as the simulation results of neutron yield, target temperature, thermal stress, and concentration of reaction byproducts, via both the hadron-based method and the photon-based method, will be presented and discussed. Moreover, the predicted performances of the two kinds of sources will be compared. The design of a high yield photoneutron source target station, aiming for potential industrial or medical applications, driven by an energetic electron linear accelerator (e-LINAC), will also be presented.

## 2. Principles of Neutron Production

### 2.1. The Reactions for Neutron Production

#### 2.1.1. Hadron-Based Method

The coulomb barrier formed by the incident nucleus and the target nucleus requires that the atomic number of the target nucleus should not be high, in the case that our motivation is to build a compact neutron source that should use low energy accelerators, rather than a spallation neutron source. As the height of the coulomb barrier is about 1.2Z_1_Z_2_(1 + 1.26Z_2_^1/3^)^−1^ (MeV), where Z_1_ is the charge of the incident charged particle and Z_2_ is that of the target nucleus, in order to ensure that the coulomb barrier can be successfully surpassed, a singly charged particle, such as proton or deuteron, is preferred to be chosen as the incident particle. The target nucleus should be with low atomic number for the easy entering of the incident charged hadron into the target nucleus. ^2^H, ^3^H, ^9^Be, and ^7^Li are the usually used target nuclei.

Once the incident hadron enters the target nucleus, an intermediate state, with the excited state energy equals to the summation of the relative kinetic energy of the incident hadron and the binding energy of the two nuclei, may exist and is followed by the emission of neutrons, carrying partial excited energy of the intermediate state. The yield of the neutron production reaction relies on the energy of the incident hadron and the type of the target nucleus, as well as the mass thickness and isotopic concentration of the target. Due to the competing of the ionization between the incident charged hadron and the orbital electrons of the target atoms, only a very small fraction of incident hadrons has the chance to enter the target nucleus before substantially losing their energy. Therefore, the maximum achievable reaction yield, even with a very thick target, would be far smaller than unity and actually is unfortunately usually smaller than 8%. The detailed calculations, with the aid of Geant4 code (release 11.0) [8], give the yield of each reaction for thick targets of different materials, as shown in Table 1.

#### 2.1.2. Photon-Based Method

Unlike the production of neutrons with the hadron-based method, two steps, but not one step, are involved in the photon-based method. The first step in the photon-based method is to convert energetic electrons to forward travelling bremsstrahlung photons, and the second step is to convert bremsstrahlung photons to photoneutrons, with the aid of a photon-to-neutron convertor whose target nuclei absorb the incident photons (mainly) via the GDR mechanism or QD mechanism. The conversion from energetic electrons to bremsstrahlung photons could be very efficient, with the yield of bremsstrahlung photons about unity, provided a high-Z target, such as tungsten or tantalum, is used for the first step conversion. The photonuclear reaction, which is the second step, however, suffers from the competing of the photoatomic reaction. The comparison between the cross sections of the photoneutron reaction and that of the photoatomic reactions indicates that the maximum reaction yield (neutron per photon) of photoneutron would be 4% or less (smaller than that of the hadron-based method), and is not sensitive to Z, the atomic number of the target nuclei, because both the photoatomic and photonuclear reactions are approximately proportional to Z for the photons of several MeV or a little higher. Typically, a high Z target should be used to realize a more compact photoneutron source.

### 2.2. The Reaction Yield and the Yield of the Neutron Source

Figure 1 demonstrates the reaction yields and the yields of neutron sources for (e→γ,n) reaction and (p,n) reaction, respectively. It could be noticed that with the increasing energies of electrons or protons, both the reaction yields for (e→γ,n) reaction and (p,n) reaction would grow and gradually achieve a saturation when their energies approach 50 MeV. For the same energy of the incident particles, the reaction yield for (p,n) reaction is about three times higher than that of (e→γ,n) reaction. Consequently, as the neutron yield of an accelerator driven neutron source is given by the product of the reaction yield (n/incident particle) and the current of the accelerator (incident particles/s), less power would be deposited onto the target for (p,n) reaction than for (e→γ,n) reaction. To achieve a neutron yield of 10^14^ n/s, a photoneutron source of 50 MeV/50 kW or a proton source of 20 MeV/21 kW should be used.

### 2.3. Short Term Consideration: The Power Deposition and Heat Dissipation

#### 2.3.1. Hadron-Based Method

Although the results shown in Figure 1 imply that the (p,n) reaction might be a more efficient way to produce neutrons, this kind of source immediately has to face the problem of heat dissipation. As shown in Figure 2, the low energy hadrons are not minimum ionizing particles (MIPs), with a specific energy loss 10 folds higher than the GeV protons used in spallation neutron sources. The very limited range of low energy hadrons in the matter, e.g., 0.5165 g/cm^2^ for 20 MeV protons, leads to the power of 21 kW for protons (for generating 10^14^ n/s) being deposited within a thickness of 2.5 mm for beryllium. Moreover, for a typical diameter of 5 cm beryllium target, the energy density would be 4.3 kW/cm^3^. Therefore, an effective way to remove the deposited heat should be provided before the (p,n) source can deliver intense neutron beams. 

There are two ways to address the problem of heat dissipation. The first way is to reduce the deposited energy density by enlarging the effective volume of target, via rotating target or applying a liquid metal target that is placed outside the accelerator [9]. If both the aforementioned targets should be placed outside the accelerator, then the energy loss of hadrons penetrating the window, which separates the vacuum inside the accelerator and the outer atmosphere, will lead to an unwanted reduction of neutron yield (~6% for 20 MeV p-Be reaction, and even higher reduction for lower energy). The second way is to enhance the capability of the cooling system. By introducing the structure of micro-channel cooling, both the interface area and the heat transfer coefficient can be drastically enlarged, with the typical figure of a 100 cm^2^ area (for an 8 cm × 8 cm heat sink) and ~10^4^ W m^−2^ K^−1^ for them, respectively. Therefore, the cooling capacity of 1–3.5 kW/cm^2^ could be achieved.

#### 2.3.2. Photon-Based Method

The scenario in the case of photon-based method is different. As there is an approximately linear relationship between the atomic number and the cross section of (γ,n) reaction, high-Z materials are usually used to convert photons to neutrons. The ~7 MeV threshold of the (γ,n) reaction for high-Z nuclides requires that the e-LINAC should deliver electrons of >7 MeV in order to generate bremsstrahlung photons with the endpoint energy of >7 MeV. Such an energy is significantly larger than 0.511 MeV, the rest mass of the electron. Thus, the electrons involved in the photon-based method must be MIPs and have the relatively large range in the matter (for example, 3.9 mm in tungsten for 50 MeV electrons). The inherent high penetrating capability of both the energetic electrons and bremsstrahlung photons enables the presence of a large volume photon-to-neutron-convertor placed outside the accelerator. A 50-μm-Ti foil is set as the window of the e-LINAC to maintain its vacuum. As the deposited energy by each penetrating 50 MeV electron is merely 30 keV, the heat dissipation of 50 MeV/50 kW electrons inside the 50-μm-Ti foil would be expected as merely 30 W, which could be deemed as ignorable. After penetrating the 50-μm-Ti foil, the transmitted electrons will be stopped within a tungsten e→γ→n convertor. The thickness of the tungsten convertor could be determined by considering both the 50 MeV electrons’ range, R_e-_ (~3.9 mm) and the mean free path, λ, of bremsstrahlung photons in tungsten, respectively. As λ is usually larger than 7 mm, 1.75 folds larger than R_e-_. A tungsten convertor with the thickness of 21 mm, which is 3λ, can be set to absorb most of the forward-emitted-bremsstrahlung-photons and convert ~1% of them to neutrons. Because of the enlarged volume of the centimeter scale tungsten target, the severe heat dissipation problem, which is present in the hadron-based method, might be significantly alleviated, although an even higher electron power (about 1.4 folds higher than 20 MeV protons for the 50 MeV electrons) is needed to achieve the same neutron yield.

#### 2.3.3. Estimation of the Neutron Yield Affected by the Heat Dissipation

For both the hadron- and photon-based methods, the sources’ neutron yields are proportional to the neutron convertor’s deposited power, the upper limit of which is determined as:(1)P=H⋅S⋅ΔT
where *P* (unit: W) is the maximum heat power that can be dissipated from the convertor; *H* (unit: W m^−2^ K^−1^) is the heat transfer coefficient of the cooling system that is affected by the material and geometrical structure of the convertor, in which the coolant flows and carries the deposited heat away; *S* (unit: m^2^) is the area of the interface where the coolant carries the deposited heat away from the convertor. By applying the micro-channel structure, not only can a large *S* be acquired, but a large *H* will also be achieved because the hydraulic diameter will be accordingly lowered and in turn increases the *H* [10]. Δ*T* (unit: K) is the wall adjacent temperature, which stands for the temperature difference between the coolant and the convertor at the interface of the cooling system. 

The typical designs of a proton and a photon-based neutron convertor are multilayer structure [11] and multidisc structure [12], respectively. The difference in *S* for the two convertors is caused by their volumes and the geometry of cooling channels. For convertors of 5 cm diameter, the *S* is 100 cm^2^ and 400 cm^2^ for proton- and photon-convertor, respectively. The Δ*T* can be routinely set as 40 K, for the typical flow velocity of several meters per second.

The upper limit of *P* in Equation (1) for proton- and photon-convertor are estimated as 16 kW and 64 kW, respectively. According to the results shown in Figure 1, the neutron yields for these two sources are 7.61 × 10^13^ n/s and 1.27 × 10^14^ n/s, respectively.

To further refine the estimation of the upper thermal limit that the convertor can withstand, 3D finite-element simulation using Ansys Workbench 2021 R2 was conducted for these two typical designs of neutron targets, as shown in Figure 3. The following requirements should be met before the upper limit of the bombarding power is approached. First, cooling water’s temperature should be lower than the boiling point, typically around 20 °C. Second, the temperature of the interface between the target and the cooling water should not be higher than the boiling point. Third, thermal stress caused by the gradience of temperature inside the convertor should not exceed the materials’ yield strength. Restrained by these requirements, simulation results indicate that the upper power limit for a 20 MeV proton-accelerator target and a 50 MeV electron-accelerator target are approximately 10 kW (0.5 kW/cm^2^) and 60 kW (4.7 kW/cm^2^), respectively. 

In summary, it can be noticed that although the reaction yield for the photon-based method (0.016 n per 50 MeV electron) is relatively smaller than that of the proton-based method (0.069 n per 50 MeV proton), the maximum achievable neutron yield for the photoneutron source is higher than that of the proton-based neutron source. This reversal is directly related with the high penetrating capability of energetic electrons and photons, which enables a large volume and then a large interface area for the tungsten convertor.

### 2.4. Long Term Consideration: Light Nuclides Production along with the Emission of Neutrons

By setting the power of bombarding hadrons or electrons within a suitable region, the related thermal stress can be well controlled under the yield strength of the convertor materials, ensuring that the convertor will not be broken by the bombarding charged particles within a short operating duration. Another issue related with the hydrogen or helium embrittlement, however, might exist and affect the long-term operating stability of the neutron source.

#### 2.4.1. Hadron-Based Method

As mentioned above, the low energy incident hadron asks for the low-Z target for the easy penetrating of coulomb barrier. However, the lowered coulomb barrier also facilitates the emission of charged particles through the exit reaction channels, which might be closed for a heavier target nucleus with the higher coulomb barrier. A Geant4 aided simulation was conducted to calculate the reaction yield of various light nuclides, such as ^1^H, ^2^H, and α. The ratios between the reaction yields for these light nuclides to that for neutrons can be deduced and shown in Table 2, by observing which it could be noticed that the elements of hydrogen or helium will be abundant in the thin-layer-convertor. Along with the increment of the integrated number of bombarding hadrons in the convertor, the elemental concentration of hydrogen and helium will also increase accordingly. The accumulation of atoms of hydrogen or helium inside the convertor may result in the blistering problem, leading to the failure of the convertor [13]. 

Usually, the incident protons are notorious for leading to the blistering problems inside the convertor, and this problem perhaps could be addressed by adding blistering layer [11] or adding water beam dump [14] behind the convertor. However, the light nuclides production, as shown in Table 2, might be a severe intrinsic problem that can hardly be addressed. Compared to helium atoms, the transmutation hydrogen atoms are mobile, and their reaction yield is relatively low. However, the produced helium atoms, which have a relative larger reaction yield, are unfortunately immobile and should be majorly concerned. As it is hard to define an accurate upper limit for helium concentration inside the beryllium convertor, merely the radiation induced swelling [15] is preliminarily considered in this study. Moreover, the relationship that the beryllium volume expansion is proportional to the helium concentration, which is summarized by Bojanowski [16], is adopted for predicting the swelling strain induced by the proton bombarding. By assuming that the swelling is isotropic and setting the constraint that the 1st principal stress should be lower than ultimate tensile strength, the critical helium concentration, $C_{He}$, thus could be calculated according to Equation (2) as 3088 appm.
(2)$$\begin{array}{cc} \;\frac{\Delta V}{V_{0}}\left(\%\right)= & 1.14\times 10^{-4}\times C_{He}\left(\mathrm{appm}\right)\\ \varepsilon_{i}= & \frac{\Delta V}{3V_{0}}\left(i=x,y,z\right)\\ \sigma_{i}= & \frac{E}{1-2\nu }\varepsilon_{i}\leq UTS \end{array}$$
where *ε* is the swelling strain and *σ* is the corresponding stress. Figure 4 shows the CERN FLUKA 4-2.2 (https://fluka.cern) [17,18] simulated helium concentration depth profile for a 20 MeV/21 kW proton source after a 1400-h-operation. The lifetime for the beryllium target is estimated to about 1400 h, when the accumulated helium concentration reaches 3088 appm. This lifetime might be further reluctantly, drastically reduced when the bombarding of protons on the target is not uniform, which has been observed by different groups although not formally reported and systematically discussed.

#### 2.4.2. Photon-Based Method

Taking the advantage of higher photoneutron cross section with increased atomic number, the alleviation of the blistering issue discussed in the hadron-based method might benefit from the applying of high-Z electron-to-photon-to-neutron convertor (EPNC). There are three reasons to support this argument. First, the height of the coulomb barrier is significantly enlarged due to the high atomic number of the bombarded nucleus, which in fact narrows the width of the exit channels for light nuclides. The peak energy of the GDR curve, which shows a dependance of E_GDR_∝A^−1/6^ with the Goldhaber–Teller (G-T) mode [19] or E_GDR_∝A^−1/3^ with the Steinwedel–Jensen (S-J) mode [20], could further help confine the light nuclides within the target nucleus, due to the even lower escaping kinetic energy of light nuclides and the even higher coulomb barrier. Second, the enlarged geometrical size of the EPNC could effectively reduce the density of hydrogen or helium atoms with the same integrated neutron yield. Third, unlike the convertor in the hadron-based method which must perform the roles of both neutron production and the vacuum insulation, the EPNC merely acts the role of neutron production and does not need to withstand the pressure between the vacuum inside the accelerator and the outer environment. The presence of a rotating or moveable EPNC for elongating the lifetime of the neutron source is thus possible.

The data presented in Table 2 also show the reaction yields of ^1^H, ^2^H, and α for the photon-based method, which are so small that their effect on the lifetime of EPNC can be ignored. For the same reason, we omit the discussion of the light nuclides production within the material of the window which separates the accelerator vacuum from the outer environment.

## 3. The Physical Design of a 50 MeV e-LINAC Based Photoneutron Target-Moderator-Reflector (TMR)

### 3.1. Target Layout and Thermal Hydraulic Simulation

To effectively cool the target, the photoneutron target is segmented into 6 square sheets of 60 mm side length and various thicknesses (shown in Figure 5). A 1.2-mm-thick gap is left between each neighboring sheets to permit the cooling water flowing through it to remove the deposited heat. This scheme has proven to meet the cooling requirement of some spallation neutron target [21] or high-power photoneutron target [22] when the heat load is on the order of 100 kW. 

The thicknesses of the sheets are 2.5 mm, 2.5 mm, 3.0 mm, 4.0 mm, 6.0 mm, and 8.0 mm individually. Because of the higher deposited energy density in the first two sheets, the smallest thickness is set for them. Tungsten is selected as target material for its high neutron yield and excellent thermal properties. To avoid fluid corrosion, each tungsten sheet is cladded with a layer of 0.25-mm-thick tantalum [23]. The tungsten/tantalum convertor, as well as the cooling water, are encapsulated within a 2-mm-steel casing, for which a 50-μm-Ti foil acting as the entrance window for the incident electrons.

Thermal hydraulics analyses are conducted by Ansys software. The realizable k-ε model and Coupled algorithm are adopted. The 3D energy deposition matrix is simulated by FLUKA and then imported to Ansys Static Thermal unit as heat generation source. The cooling water enters the convertor from the front surface, where the electrons enter, in order to help the sheets, which are close to the entrance surface of the convertor, remove the deposited heat because they deposit more energy. The flow cross section is intentionally set gradually narrower for the back sheets, in order to alleviate the recirculation problem in outlet channel to some extent. With 3.6 L/s inlet velocity of the cooling water, the maximum temperature for a 50 kW target can be maintained under 114.61 °C. The contacts between target sheets and steel casing are set as frictional contacts, and the water pressure is 3 bar. Simulation results presents that the maximum thermal stress of tungsten sheets is 253.3 MPa, which is lower than the yield strength of tungsten. The maximum water temperature is approximately 100 °C, which leaves a margin for water boiling temperature, 133.4 °C at 3 bar. Finally, the predicted neutron yield of the target is simulated as 8.125 × 10^13^ n/s.

### 3.2. Moderation and Shielding of Neutrons

To carry out the slow neutron applications, the fast photoneutrons should be slowed down with a TMR system. The moderation process should be realized effectively so that the flux of thermalized neutrons is as large as possible. 

The structure of the TMR system in this work is shown in Figure 6, in which a polythene cylinder is placed to surround the tungsten convertor. As the size of tungsten convertor is selected as 6 cm × 6 cm × 4.15 cm, photoneutrons produced by the (γ,n) reaction will firstly lose their kinetic energies via the inelastic scattering with the tungsten nuclei. Once the photoneutrons leave the tungsten convertor, they subsequently undergo the slowing down process mainly via the elastic scattering with protons in the polythene. Outside the polythene cylinder is the cylinder of graphite, which reflects neutrons escaping from the polythene cylinder back to it. A 20-cm-thick lead layer surrounds the graphite layer to absorb all the bremsstrahlung photons, inelastic scattering gamma-rays and radiative capture gamma-rays along with the production and moderation of photoneutrons. The layer of boron doped polythene outside the lead layer will not act the role of reflecting the possible leaking neutrons back to the inside layers, rather slowing down and absorbing them. An additional 10-cm-thick lead layer is placed outside the boron doped polythene to absorb the radiative capture gammas-rays produced in it. Finally, a 100-cm-thick heavy concrete is used to help reach a boundary dose rate lower than 2.5 μSv/h, ensuring radioactive safety for the public. 

A FLUKA simulation was conducted to track the fluxes and spectra of photons and neutrons inside the TMR. The simulated results help acquire a contour plot of dose rate for both the neutrons and photons, respectively, as shown in Figure 7. It can be seen that a 150 cm radius might be adequate to reach the 2.5 μSv/h dose rate limit. 

### 3.3. Tailoring the Ratio between the Bremsstrahlung Photons and the Emitted Neutrons

One of the major arguments of the photoneutron source might be the intense photon background accompanying the production of neutrons. However, benefiting from the pulsed mode of the e-LINAC with a typical pulse width of several microseconds and a repetition frequency of hundreds Hz, the detection of slow neutrons can be readily conducted by selecting a measuring temporal duration free of the interference of the X-rays flash. Moreover, as both the X-rays and neutrons are produced within the same system, a bi-modal imaging is possible by fusing the X-ray imaging and neutron imaging together, as we have shown [24]. In the case that the intensity of X-ray pulses should be attenuated while that of the neutron should be kept, a filter that is nearly transparent to slow neutrons while opaque to X-rays should be used. Bismuth is perhaps the best candidate for the large cross sections for photons due to the high atomic number and the low absorption cross section for slow neutrons due to the magic number of neutrons for ^209^Bi. A simulation, as shown by Figure 8 and Figure 9, was conducted to show the performance of tailoring the emitted spectra of neutrons and photons, by placing bismuth of various thicknesses at the entrance of the channel via which neutrons will fly along the direction heading toward the detector. 

A neutron channel, which has a length of ~2 m and sectional area of 10 cm × 10 cm, is designed for the extraction of thermalized neutrons. By simulation, the total thermal neutron (10–300 meV) flux at the entrance of the neutron channel is 4.34 × 10^10^ cm^−2^ s^−1^, with a peak brilliance of 1.49 × 10^10^ cm^−2^ s^−1^ sr^−1^ for neutrons heading toward the detector. Due to the existence of polythene, graphite, and lead surrounding the neutron channel, the inverse-square law for the neutron flux will not be obeyed before the travelling neutrons finally leave the neutron channel. Therefore, the brilliance of thermalized neutrons at the exit of the neutron channel needs to be simulated for predicting the flux of collimated neutrons at the position of the imaging neutron detector, which might be placed 10 m or more away for a 100:1 or even larger imaging aspect ratio. As illustrated by Figure 10, the thermal neutron flux is approximately isotropic at the neutron channel entrance, while confined to be lower than ~3° at the end of the neutron channel. The simulated result shows that the brilliance at the exit of the neutron channel is 1.03 × 10^10^ cm^−2^ s^−1^ sr^−1^, indicating a thermal neutron flux of 8.09 × 10^5^ cm^−2^ s^−1^ for the imaging neutron detector placed 10 m away can be acquired. 

## 4. Discussion

Benefitting from the high penetrating capability of the energetic electrons, the EPNC can be placed outside the accelerator. Because this convertor does not perform the role of vacuum insulation, its size would not be limited by the structure of the accelerator. The relatively large free path of photons inside the convertor leads to a large volume of the convertor. Therefore, the cooling system can be optimally designed to reach as high a heat dissipation capability as possible, in order to finally achieve a high neutron yield. Moreover, a rotating or moveable EPNC can be designed to further augment the thermal capacity of the convertor. 

As shown in Figure 1, a saturation of the reaction yield can be noticed when the energy of bombarding electrons is higher than 50 MeV, indicating that increasing the energy of bombarding electrons will no longer be an effective way to increase the neutron yield. Another disadvantage along with increasing the electron’s energy is that the QD mechanism, which produces more energetic neutrons, will be more prominent. Therefore, the moderation for neutrons would be difficult and the flux for moderated neutrons would be accordingly lowered.

Besides the major problem of heat dissipation inside the tungsten target, a minor problem of heat dissipation would exist for the peripheral materials, such as the polythene moderator and graphite reflector, because the bremsstrahlung photons escaping from the tungsten target may deposit heat inside them. In order to clarify this issue, we conducted a Monte Carlo simulation and the result shows that for a 50-kW electron-accelerator-based neutron source, most of the heat (say, 44.9 kW) will be deposited inside the tungsten target, and the heat, deposited by bremsstrahlung photons, inside the polythene moderator and graphite reflector is less (say, 1.26 kW and 2.28 kW, respectively). Considering the volumes of the polythene moderator and the graphite reflector are 1.6 × 10^4^ cm^3^ and 2.27 × 10^5^ cm^3^, the heat deposition rate inside them would be 0.079 W/cm^3^ and 0.01 W/cm^3^, respectively. Both of these values are comparable with the heat deposition rate inside the polythene moderator mentioned in Hirotaku Ishikawa’s study [25], in which the heat deposition rate was not explicitly demonstrated and could be roughly estimated as 0.044 W/cm^3^, assuming that the heat deposition in the 720 cm^3^ polythene is 1/40 (the e-LINAC’s power is ~1.224 kW in Hirotaku Ishikawa’s study) that of our case. It must be mentioned that the result of 0.044 W/cm^3^ is an overestimation due to the smaller volume of polyethene moderator in Hirotaku Ishikawa’s study. Therefore, the temperature increase of the polythene moderator in our case would perhaps be even higher if all the other configurations are assumed as the same. To address this potential problem of moderator’s temperature increase, which may affect the neutron energy spectrum when long wavelength neutrons are the major concern, the relatively larger volume, of the polythene moderator in our case, might be helpful because it enables the existence of internal cooling channels, in which the flowing cooling water also acts as the same role of slowing neutrons as the polythene moderator. Therefore, the temperature increase of the moderator in our case would be alleviated. As to the graphite reflector, the temperature increase problem would not be severe because of the lower heat deposition rate (0.01 W/cm^3^) and high thermal conductivity, 168 W/(m K) of graphite, about two orders of magnitude higher than that of polythene, 0.42 W/(m K) [26]. A detailed design of the internal cooling channels inside the polythene moderator, as well as the temperature increase both in the polythene moderator and graphite reflector, will be conducted in a future study.

## 5. Conclusions

As MIPs, energetic electrons only lose ignorable kinetic energy when they penetrate a thin titanium foil, which may act as a window for separating the outer atmosphere and vacuum inside the accelerator. Therefore, an isolated EPNC can be set outside for an e-LINAC to enable the production of photoneutrons, which subsequently need an effective moderation process to be slowed down to wanted slow neutrons. Although the photoneutron reaction yield is relatively smaller than that of the hadron-based method, the larger size of the EPNC significantly elevates its maximum deposited heat power, which in turn drastically improves the total neutron yield of the neutron source. Besides, the relationship whereby the σ_GDP_ ∝ Z asks for a high-Z photon-to-neutron convertor, might bring at least two advantages: (1) the lowered peak of σ_GDP_ curve means that the energy of photoneutrons produced with high-z target is also lowered, resulting in a more effective moderation of fast neutrons; (2) the high coulomb barrier of the high-Z nuclei of the convertor, hinders the emission of light nuclides from the compound nucleus when the target nucleus absorbs an incident photon, thus alleviating the problem of hydrogen/helium embrittlement, which would be intrinsically severe for the hadron-based method, and practically could be neglected for the photon-based method. We therefore might draw the conclusion that, in order to realize an intense low energy accelerator driven neutron source, a photoneutron source would be one of the best candidates able to strike a compromise among the neutron yield, long lifespan, spatial footprint, and even cost-effectiveness.

A conceptual design of the TMR system for a 50 MeV/50 kW e-LINAC is proposed and presented. The total neutron yield of 8.125 × 10^13^ n/s, and epithermal neutron’s (0.5 eV–10 keV) flux of 4.56 × 10^10^ cm^−2^ s^−1^ at the entrance of the neutron channel, as well as the thermalized neutrons’ brilliance of 1.03 × 10^10^ cm^−2^ s^−1^ sr^−1^ at the exit of the neutron channel, are very promising to permit various neutron applications, including BNCT treatment or rapid neutron imaging to be conducted. Moreover, the boundary dose rate can be controlled under 2.5 μSv/h to ensure the radioactive safety.

## Figures and Tables

**Figure 1 materials-15-07674-f001:**
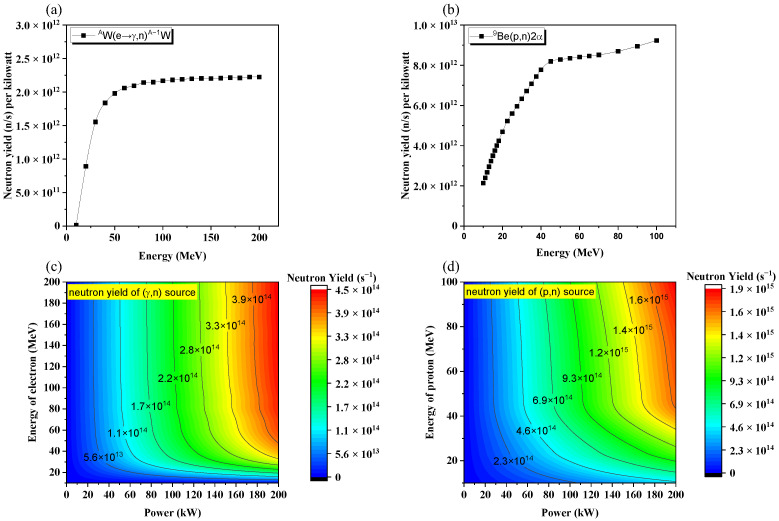
(**a**) The (e→γ,n) reaction yield with tungsten target; (**b**) the (p,n) reaction yield with beryllium target; (**c**) the yield of a neutron source with different energies and powers of incident electrons; (**d**) the yield of a neutron source with different energies and powers of incident protons.

**Figure 2 materials-15-07674-f002:**
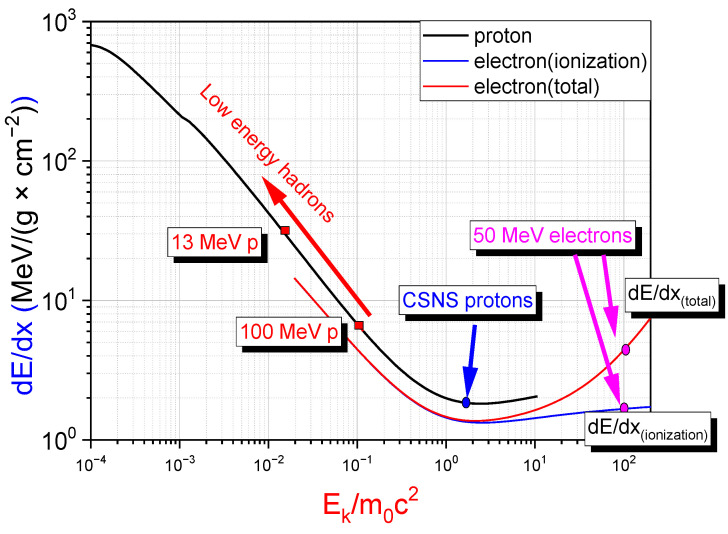
The specific energy loss of different particles with different energies in the matter.

**Figure 3 materials-15-07674-f003:**
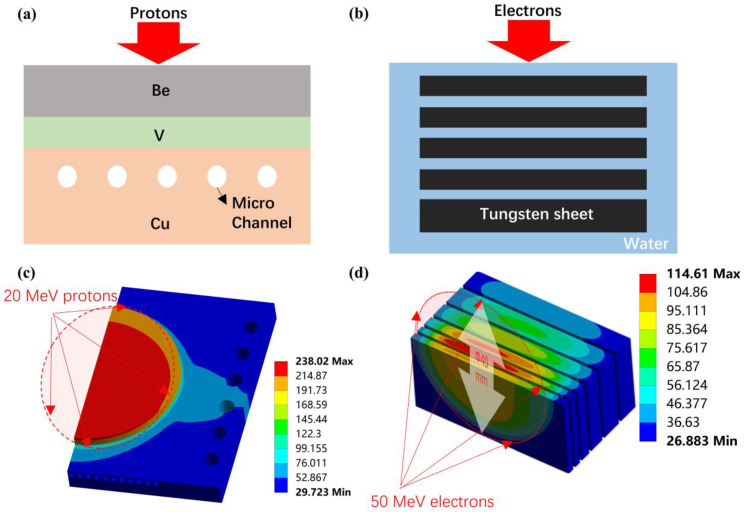
The typical target design: (**a**) multilayer structure for proton and (**b**) multidisc structure for electron; (**c**) the result of the 3D thermal-hydraulic simulation for 20 MeV proton-accelerator target and (**d**) 50 MeV electron-accelerator target.

**Figure 4 materials-15-07674-f004:**
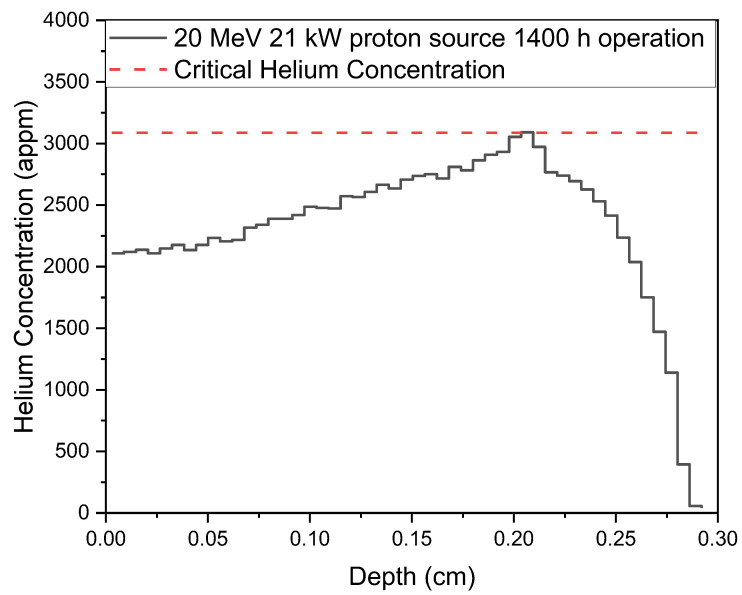
The concentration distribution of helium atoms generated by the 20 MeV/21 kW proton source after a 1400-h-operation.

**Figure 5 materials-15-07674-f005:**
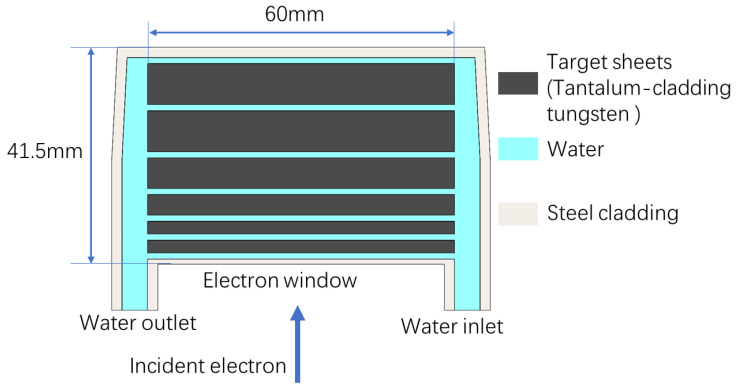
The schematic diagram of the neutron target.

**Figure 6 materials-15-07674-f006:**
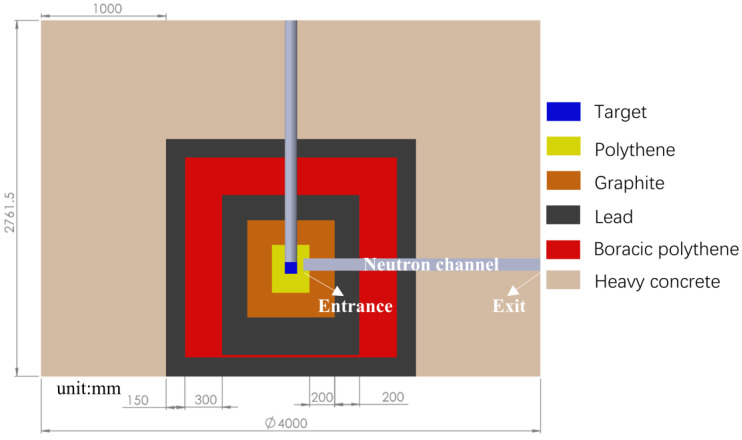
The geometrical arrangement of the multilayer structure TMR.

**Figure 7 materials-15-07674-f007:**
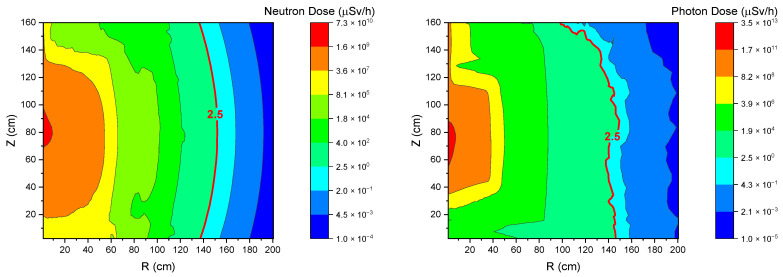
Neutron and photon dose map for TMR.

**Figure 8 materials-15-07674-f008:**
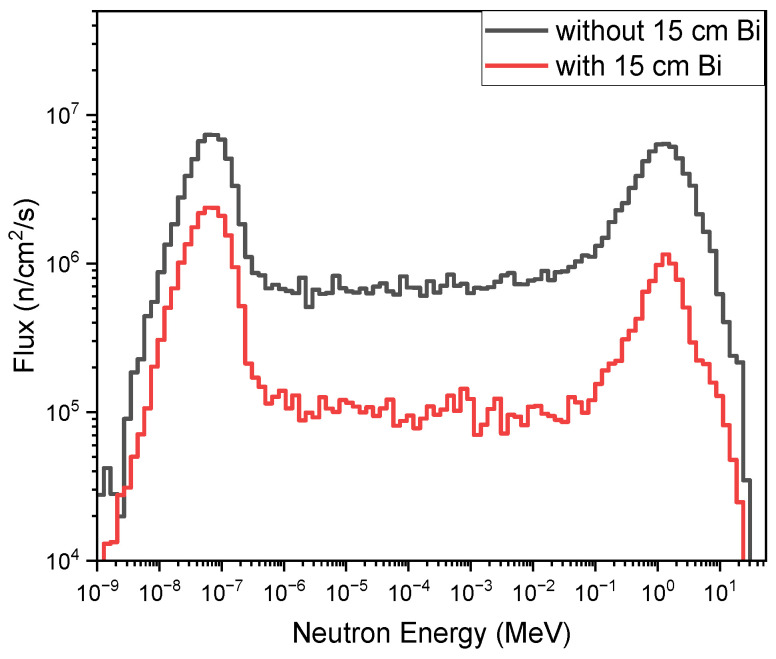
Neutron spectra at the end of the exit channel with or without 15 cm bismuth.

**Figure 9 materials-15-07674-f009:**
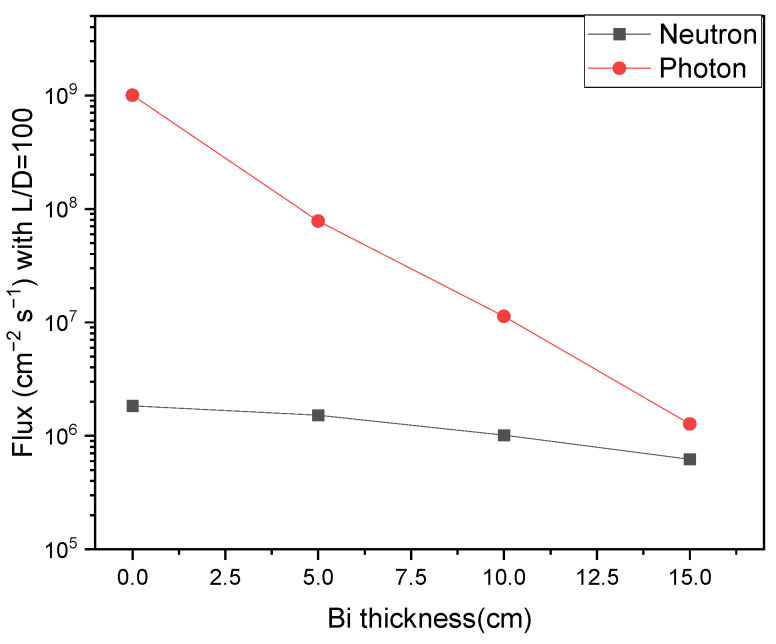
The variance of neutron and photon flux with different thicknesses of bismuth at 10 m away from the exit of the neutron channel (L/D = 100).

**Figure 10 materials-15-07674-f010:**
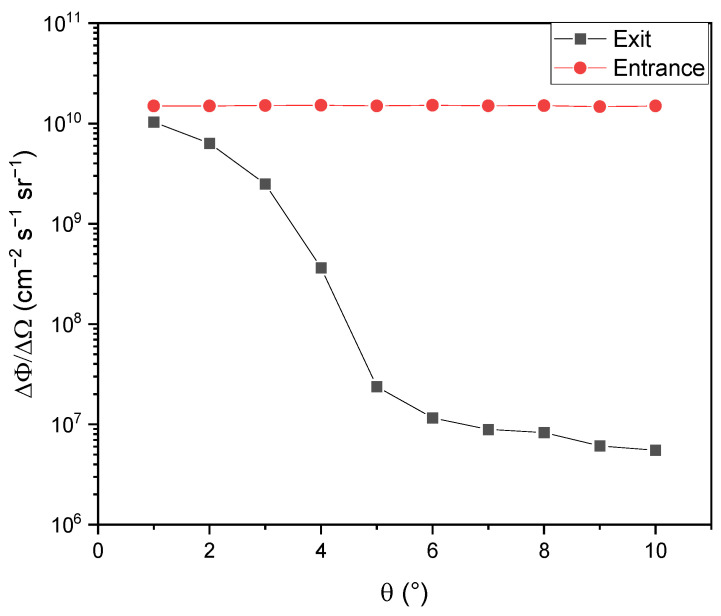
Thermal neutron brilliance at the entrance and exit of the neutron channel.

**Table 1 materials-15-07674-t001:** Reaction yields of different targets for incident electron and proton of various energies.

Incident Particle	Target	Reaction Yield (Neutron per Incident Particle)
20 MeV e^−^	^nat^W	2.85 × 10^−3^
50 MeV e^−^	^nat^W	1.58 × 10^−2^
100 MeV e^−^	^nat^W	3.47 × 10^−2^
100 MeV e^−^	^nat^U	7.16 × 10^−2^
10 MeV p	^nat^Li	3.26 × 10^−3^
10 MeV p	^9^Be	3.44 × 10^−3^
20 MeV p	^9^Be	1.52 × 10^−2^
50 MeV p	^9^Be	6.89 × 10^−2^
50 MeV p	^181^Ta	7.71 × 10^−2^

**Table 2 materials-15-07674-t002:** The reaction yields of ^1^H, ^2^H and α for proton-accelerator and electron-accelerator neutron target.

	Light Nuclides	^1^H	^2^H	α	Estimated Lifetime
Incident Particles		Ratio of Light Nuclide Reaction Yields to That of the Generated Neutrons (1/n)			
20 MeV p (^9^Be target)		0.75	0.30	2.5	1400 h
50 MeV e^−^ (^nat^W target)		3.5 × 10^−4^	1.5 × 10^−5^	2.1 × 10^−6^	Very long

## Data Availability

The data presented in this study are available on request from the corresponding author.

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
