# Peer review of "A Design for the High Yield Photoneutron Source Target Station"

_materials, 2022, doi:10.3390/ma15217674_

Round 1

Reviewer 1 Report

Your work is original and sound. It should be published after tiny corrections that are pointed out in the attached file (Reviewer report.pdf)

Author Response

Thanks for the reivewer's nice comments. Please see the attachment.

Reviewer 2 Report

This manuscript describes a conceptual design of a low-energy accelerator-driven neutron source. I believe this manuscript is out of the scope of this journal, Materials. A neutron source may contribute to the advancement of material characterization techniques, however, it requires many engineering steps to realize the neutron source described in the manuscript. Therefore, I recommend considering submitting this paper to other journals.

Before resubmission, the authors need to fix error messages of reference sources and misuses of capital/small letters.

Author Response

(The authors gave the same response as above.)

Reviewer 3 Report

There are a large number of “Error! Reference source not found.” occurrences in the document that require correction.

Line 25: “have been proved to be that can play” should be “have proved to play”

Line 143: “may be” should be “is”

Line 191: suggest that ~1/100 be changed to ~1%

Line 299: define G-T mode and S-J mode

Line 370: “Figure 1” should be “Figure 7”

Line 385: should this refer to Figure 8 and some other figure?

Line 405: This should be Figure 8

Line 417: delete “an”

Line 439: “leads” is the wrong word. Maybe “means”?

Author Response

(The authors gave the same response as above.)

Round 2

Reviewer 2 Report

I understand the situation of the special issue. It may be better to comment that a heat removal function may be required for the polythene moderator for those who advance this neutron source project. A temperature increase of a polythene moderator is reported to be about 5 K within one hour in the case of a 1-kW class electron-accelerator-based neutron source. It suggests 50-kW neutron source needs cooling channels not only in a converter but also in other components.

Author Response

We thanks to the reviewers' nice advice. Please see the attachment for the point-by-point response.

As to the manuscript, we provide the following modifications:

  • Add a new paragraph in Discussion section “Besides the major problem of heat dissipation ……would be conducted in the future study” to response to reviewer2’s comments.
  • Add two citations, citation 25 and 26, with the former one referring to Hirotaku Ishikawa’s work in HUNS and the latter one referring to the thermal conductivity data source.
  • In section 3.1, “… a layer of 0.5-mm-thick tantalum …” is corrected as “… a layer of 0.25-mm-thick tantalum …”.

Thank you for the consideration.

Yigang Yang.
